# Chemical Modifications Influence the Number of siRNA Molecules Adsorbed on Gold Nanoparticles and the Efficiency of Downregulation of a Target Protein

**DOI:** 10.3390/nano12244450

**Published:** 2022-12-14

**Authors:** Anna V. Epanchintseva, Julia E. Poletaeva, Anton S. Dome, Ilya S. Dovydenko, Inna A. Pyshnaya, Elena I. Ryabchikova

**Affiliations:** Institute of Chemical Biology and Fundamental Medicine SB RAS, Novosibirsk 630090, Russia

**Keywords:** modified siRNA, phosphoryl guanidine modification, siRNA adsorption on AuNPs, multilevel nanoconstruct, efficiency of silencing

## Abstract

Small interfering RNAs (siRNAs) are a powerful tool for specific suppression of protein synthesis in the cell, and this determines the attractiveness of siRNAs as a drug. Low resistance of siRNA to nucleases and inability to enter into target cells are the most crucial issues in developing siRNA-based therapy. To face this challenge, we designed multilayer nanoconstruct (MLNC) with AuNP core bearing chemically modified siRNAs. We applied chemical modifications 2′-OMe and 2′-F substitutions as well as their combinations with phosphoryl guanidine group in the internucleotide phosphate. The effect of modification on the efficiency of siRNA loading into nanocarriers was examined. The introduction of the internucleotide modifications into at least one of the strands raised the efficiency of siRNA adsorption on the surface of gold core. We also tested the stability of modified siRNA adsorbed on gold core in the presence of serum. Based on loading efficiency and stability, MLNCs with the most siRNA effective cargo were selected, and they showed an increase in biological activity compared to control MLNCs. Our study demonstrated the effect of chemical modifications of siRNA on its binding to the AuNP-based carrier, which directly affects the efficiency of target protein expression inhibition.

## 1. Introduction

Small interfering RNAs (siRNAs) are currently recognized as the most effective means of influencing protein synthesis in the cell and are investigated as one of the promising options for gene therapy of various diseases, including cancer [1,2,3,4,5]. Nonetheless, among the siRNA drugs approved for clinical use, there is not a single one aimed at treating cancer. This situation is due to several problems (associated with complex structure of tumors, biological properties of siRNA, and the human body’s response to RNA therapy) that have not yet been resolved. A detailed analysis of approaches to solving these problems is presented in recent reviews [1,2,4,6]. The creation of “good” carriers (vehicles for siRNA)—which can not only deliver it to the cell but also prevent many accompanying undesirable events—is one productive approach.

Thousands of research articles have been published about the creation of various carriers of nucleic acids, including siRNA, on the basis of nanoparticles of various types. An exhaustive overview of the current state of development of siRNA carriers is presented in several reviews [1,4,7,8]. Among known siRNA carriers, those mostly based on lipids predominate and facilitate the entry of siRNA into the cell. By structure, lipid carriers are categorized into liposomes, stealth liposomes, solid lipid nanoparticles, lipoplexes, stable nucleic acid lipid particles, and lipid nanoparticles [2,6,7,8,9].

There is a class of carriers based on metal nanoparticles to which siRNA is bound. Recent reviews [7,10] (i) provide a detailed examination of inorganic nanoparticles’ properties that justify their use for the delivery of siRNA and (ii) point to good prospects for further developments in this direction. Gold nanoparticles (AuNPs) carrying covalently bound siRNAs (the so-called spherical siRNAs) have been approved for (and successfully passed) phase 0 clinical trials [11], suggesting that AuNPs hold promise as a component of siRNA delivery systems. For several years, we have been developing an siRNA carrier [12,13] (based on AuNPs) whose structural organization differs from all previously published ones. In this carrier, molecules of siRNA are noncovalently adsorbed onto already prepared AuNPs, thus forming a core. The cores are then covered with a lipid shell doped with an amphiphilic peptide; in other words, a multilayer nanoconstruct (MLNC) is sequentially assembled (Figure 1). The process of preparation, purification, and concentration of this MLNC is described in detail in ref. [13].

The main advantage of our approach to the creation of an MLNC is the ability to control and modify each stage of this process. This ability allows a researcher to obtain the best possible result at each stage and to ensure accurate characteristics of all intermediate products. A similar approach has been used by Shaabani et al. [14] to obtain an siRNA carrier: they coat AuNPs (obtained in the presence of chitosan) with a layer of siRNA and next by an outer shell consisting of chitosan. In their work, siRNA was not adsorbed directly on the surface of the AuNPs but on the chitosan layer covering the AuNPs.

Application of the step-by-step approach to the creation of an MLNC has enabled us to study in detail the physicochemical properties of the resulting core (AuNP-siRNA) and the effect of storage on its properties [15,16].

Any carrier of siRNA must ensure the “safe” circulation of siRNA in biological fluids, by protecting siRNA from adverse factors outside cells, and ensuring the penetration of siRNA into target cells and entry into the cytosol. On this path, siRNA inevitably “suffers losses”: no more than 2–3% of nanocarriers introduced into the bloodstream enter the cells, and, according to researchers, only 1–2% of siRNA molecules reach the cytosol, where their molecular targets (mRNA) are located [17,18,19,20]. Nevertheless, the highest efficiency of the “functioning” of siRNA causes the suppression of the synthesis of the target protein to some extent, and this “extent” can be increased by protecting siRNA from intracellular nucleases that attack it after its exit from the complex with the carrier and from endosomes [1,4]. One of the ways to protect siRNA from intracellular nucleases is a chemical modification of its strands that makes them inaccessible to the enzymes. The choice of chemical modifications is not trivial; in particular, it may be limited by compatibility with the delivery system used because modifications can affect the efficiency of loading of the system and, later, the release of siRNA from the carrier [8,21].

Previously, our institute has developed a new method for modifying nucleic acids in the internucleotide phosphate, where the phosphitetriester bond is transformed into a phosphoryl guanidine (PG) modification during amidophosphite synthesis of oligonucleotides [22]. Some properties of PG-based modifications (hereafter: PG modifications) have been researched. For instance, the presence of a PG modification in oligonucleotides does not block the activity of some nucleic-acid–processing enzymes, for example, RNA-dependent DNA polymerase [23]. This modification provides a high degree of protection of DNA oligonucleotides from nucleases, as demonstrated in relation to snake venom phosphodiesterase, in the presence of which PG derivatives of DNA retained integrity for up to 150 h [24].

These exciting findings inspired us to study the protective effect of the PG modification on siRNAs, which are more susceptible to degradation by nucleases than DNA is; this problem limits the practical uses of siRNAs. We synthesized a series of siRNAs carrying different combinations of PG modification and modifications at the 2′ position of ribose. We found that the introduction of a PG modification into siRNA enhances its resistance to RNase A, whereas transfection of the SC-1 cell line (stably expressing GFP) with lipoplexes of siRNA containing a sense strand with PG modifications revealed that the biological effect of the siRNA is preserved [25]. It should be emphasized that in these experiments, siRNAs were delivered into cells via Lipofectamine, which is suitable only for laboratory studies owing to its high toxicity. The experiments presented in this paper had several aims: to determine the dependence of the number of siRNA molecules adsorbed on AuNPs on chemical modifications, and to investigate the efficiency of delivery of the modified siRNAs via MLNCs based on AuNPs and a release of these siRNAs from the nanocomplex.

## 2. Materials and Methods

### 2.1. Chemicals

Tetrachloroauric acid trihydrate (HAuCl_4_·3H_2_O) was purchased from Aurat (Moscow, Russia), and RNA phosphoramidites for oligoribonucleotide synthesis were acquired from Sigma-Aldrich (Hamburg, Germany). Sodium chloride (NaCl) and magnesium sulfate (MgSO_4_) were purchased from Honeywell (Seelze, Germany), and sodium citrate dihydrate (Na_3_C_6_H_5_O_7_·2H_2_O) from Fluka (Buchs, Switzerland). Egg phosphatidylcholine and 1,2-dioleoyl-sn-glycero-3-phosphoethanolamine (DOPE) were bought from Avanti (Alabaster, AL, USA), and 2-[4-dodecylamino-6-oleylamino-1,3,5-triazine-2yl]-(2-hydroxyethyl)amino]ethanol (DOME2) was synthetized as described elsewhere [12]. Disodium phosphate dihydrate (NaH_2_PO_4_·2H_2_O) and monosodium phosphate dodecahydrate (Na_2_HPO_4_·12H_2_O) were purchased from Reatex (Moscow, Russia), and uranyl acetate from SPI (West Chester, PA, USA). Peptide NH_2_-(RL)_4_G-C(O)NH_2_·5CF_3_COOH was acquired from Diapharm (Lyubertsy, Russia). Sodium acetate trihydrate (CH_3_COONa·3H_2_O), acetic acid (CH_3_COOH), and sucrose (C_12_H_22_O_11_) were bought from Panreac (Barcelona, Spain), whereas trichloromethane (chloroform, CHCl_3_) and methanol (CH_3_OH) from Reachem (Moscow, Russia). Water was purified by means of a Simplicity 185 water purification system (Millipore, Burlington, MA, USA) and had a resistivity of 18.2 MΩ·cm at 25 °C.

### 2.2. Preparation of Core Nanoparticles (AuNP-siRNA)

The MLNC developed by us consists of a core (an AuNP noncovalently coated with siRNA) and a lipid envelope. To prepare the core, citrate-stabilized AuNPs (hereinafter AuNPs) were synthesized as described in ref. [26]. The resulting AuNPs were 12 ± 1 nm in size (according to transmission electron microscopy [TEM] data) and 17.3 ± 2.1 nm (according to dynamic light scattering [DLS] analysis) and had a zeta potential of −33.6 ± 2.0 mV (data of DLS analysis). The concentration of the AuNP suspension was 3.6 × 10^−9^ M.

We used siRNA that suppresses GFP synthesis [27]. The synthesis of this siRNA has been described in detail in our previous work [25]; the sense and antisense strands are listed in Table 1.

siRNA antisense strands (100 nmol) were 5′-[^32^P]-labeled for 2 h at 37 °C in a solution (10 μL) containing 50 mM Tris-HCl pH 7.6, 10 mM MgCl_2_, 5 mM DTT, 0.1 mCi γ-[^32^P]ATP, and 80 U of T4 polynucleotide kinase. [^32^P]-labeled RNAs were purified by electrophoresis in a 15% polyacrylamide gel containing 8 M urea.

The core nanoparticles were prepared as described before [13]. First, 7.2 × 10^−7^ M siRNA was kept at 95 °C for 3 min in the presence of 0.1 mM MgSO_4_ and 5 mM NaCl, next, cooled to 25 °C and then added (at 3.6 × 10^−9^ M) to AuNPs. The mixture was incubated for 22 h at room temperature.

The resultant suspensions of the core nanoparticles were centrifuged, the supernatant was discarded, and the pellet containing the core nanoparticles was washed with 1 mL of 4 mM Na_3_C_6_H_5_O_7_·2H_2_O. The size and monodispersity of the obtained core nanoparticles were analyzed by TEM and DLS. The zeta potential of the core nanoparticles and the polydispersity index (PDI) of the suspension were assayed by DLS. The core nanoparticles had a surface plasmon resonance maximum at 520 nm.

### 2.3. Lipid Film (LF) Preparation

The LF was prepared as described in our previous work [13]. In brief, 90 μL of 1 mM egg phosphatidylcholine and DOPE in a CHCl_3_/CH_3_OH mixture (1:1) and 10 μL of 1 mM DOME2 in CHCl_3_ were added to 1 mL of CHCl_3_ in a 10 mL round-bottom flask. After that, the solvent was evaporated either at 25 °C or without thermostatting at 12 mmHg. The resultant LF was dried in vacuum in a desiccator to remove traces of the organic solvents, after which the flask with the film was kept at −18 °C for 16 h. All the procedures for obtaining the LF were carried out in an argon atmosphere.

### 2.4. Synthesis of a Peptide Conjugate with Stearic Acid

The conjugate of a peptide [(RL)_4_G-NH_2_] with stearic acid [Str-(RL)_4_G-NH_2_] for doping the lipid envelope was synthesized as described previously [12].

### 2.5. Assembly of the MLNC

This procedure has been described in detail in our previous work [13]. In brief, 100 µL of a core nanoparticle suspension in 4 mM Na_3_C_6_H_5_O_7_ (2.5 pmol in terms of gold) was mixed with a solution (0.9 mL of H_2_O and 31 µL of NaH_2_PO_4_, 0.01 M, pH 4.5) and applied onto the LF surface. The system was sonicated for 15 min (25 °C, 90 W). Then, 69 µL of 0.01 M Na_2_HPO_4_ was added to the mixture, after which 10 µL of 1 mM stearic-acid-conjugated peptide, Str-(RL)_4_G-NH_2_, was introduced, and the reaction mixture was sonicated (90 W) for 5 min at 25 °C. The size and monodispersity of the obtained MLNCs were verified by TEM and DLS. The MLNCs had a surface plasmon resonance maximum at 534 nm. The suspension of MLNCs had a reddish-pink color.

Purification of the MLNCs was carried out in solutions of sucrose, according to ref. [13]. In a 15 mL test tube, we layered 1000 µL of the MLNC suspension on top of 10 mL of an aqueous 58% sucrose solution in 1 mM phosphate buffer pH 7.2 (ρ = 1.267 g/mL, µ = 42.8 cP), followed by centrifugation at 25 °C, 2000× *g* for 15 min. The middle of the colored fractions was collected into a separate 1.5 mL tube and purified to remove excess sucrose by centrifugation at 1000× *g* and 25 °C for 10 min. The resulting preparation of MLNCs was analyzed via DLS, TEM, and UV-Vis spectroscopy.

To evaluate siRNA loading capacity (the number of siRNA molecules per one AuNP) on the basis of radioactivity, the following Equation (1) was employed:n = [siRNAb]/[AuNP0](1)
where n is siRNA loading capacity, [siRNAb] is the concentration of siRNA adsorbed on AuNPs, and [AuNP0] is the total concentration of AuNPs.

### 2.6. Assessment of MLNC Resistance to Nucleases

To examine resistance to serum nucleases, MLNCs containing the radiolabeled siRNA’s antisense strand were incubated at 37 °C for 4 h with a 10% FBS solution in DMEM. After that, the mixtures were washed with 4 mM Na_3_C_6_H_5_O_7_ and then centrifuged (30 min, 13,200× *g* rpm, 25 °C). The supernatant was analyzed by polyacrylamide gel electrophoresis (15% gel). The electrophoresis was conducted in TBE (89 mM Tris, 89 mM H_3_BO_3_, 2 mM EDTA, pH 8.3) for 90 min at 20 V/cm. Imaging Screen-K was exposed to the gels containing [^32^P]-labeled samples. Images were recorded with the help of a PharosFXTM Molecular Imager System (Bio-Rad, Hercules, CA, USA) and quantified in the Quantity One software (Bio-Rad, Hercules, CA, USA). Representative gel image is given in Appendix A.

### 2.7. Characterization of Nanoobjects

#### 2.7.1. Optical Extinction Spectra

These spectra of the core nanoparticles and all preparations of MLNCs were recorded on a Clariostar plate fluorimeter (BMG, Labtech, Ortenberg, Germany) in the range 400–800 nm according to the manufacturer’s instructions.

#### 2.7.2. DLS

Hydrodynamic characteristics of the core nanoparticles and all samples of MLNCs were evaluated by photon correlation spectroscopy on a Malvern Zetasizer Nano instrument (Malvern Instruments, Worcestershire, UK). The measurements were performed at least in triplicate.

#### 2.7.3. TEM

All core and MLNC samples were adsorbed on formvar-coated copper grids. An excess of the liquid was removed with a pipette, and the grid was placed on a drop of uranyl acetate for 30 s. Then, the excess of the stain was removed by filter paper, and the grid was air dried. The samples were examined under a JEM 1400 transmission electron microscope (Jeol, Tokyo, Japan) equipped with a Veleta digital camera (EM SIS, Muenster, Germany). Particle sizes were determined in the iTEM software version 5.2 (EM SIS, Muenster, Germany).

### 2.8. Cells

Cell line HEK293FT (Invitrogen, Waltham, MA, USA) was maintained in Dulbecco’s modified Eagle’s medium (DMEM) (Life Technologies, Paisley, UK) supplemented with 10% of heat-inactivated fetal bovine serum (FBS) (Life Technologies, Paisley, UK) and 1% of PeniStrep (Life Technologies, Grand Island, NY, USA). The cells were cultivated in culture plates and dishes (TPP, Trasadingen, Switzerland) at 37 °C in a humidified incubator containing 5% of CO_2_ in the atmosphere.

The HEK293FT cell line stably expressing EGFP (enhanced green fluorescent protein) was obtained by lentivirus transduction. For this purpose, virus particles were prepared first. HEK293FT cells (4 × 10^6^) were seeded in a 10 cm^2^ Petri dish and incubated for 24 h; then, the cells were transfected. For this purpose, we mixed a 0.25 M solution of CaCl_2_ (1 mL) and plasmids: pLP1 (containing genes gag and pol), pLP2 (containing gene Rev), pLP/VSVG (containing the gene of vesicular stomatitis virus glycoprotein G), and pLVX-CMV-Fluc-P2A-EGFP-PGK-Puro (containing gene EGFP), 4 µg each. The prepared mixture was incubated for 10 min at ambient temperature. Then, the mixture of the plasmids was diluted with 1 mL of HBS buffer (280 mM NaCl, 100 mM HEPES, 1.5 mM Na_2_HPO_4_, pH 7.12). The obtained solution was added drop by drop to HEK293FT cell culture. After 6 h, the transfection was stopped by the replacement of the growth medium with a fresh one. Cells were incubated for 48 h, and then viral particles were harvested. The medium containing viral particles was centrifuged for 15 min at 300× *g* and 4 °C. The supernatant was passed through a 0.45 µm filter, and aliquots (1 mL) were kept frozen at –70 °C.

For integration of the EGFP gene into the host cell genome, HEK293FT cells were seeded in a 6-well plate and grown to 70% confluence. For the transduction, the growth medium in wells was replaced with the medium containing viral particles. After 48 h, EGFP-positive HEK293FT cells were sorted on a S3e Cell Sorter (Bio-Rad, Hercules, CA, USA). These obtained EGFP-expressing HEK293FT cells (Appendix A) were used for testing the efficiency of siRNA delivery by MLNCs.

### 2.9. Flow Cytometry

Efficiency of EGFP gene silencing after MLNC transfection was quantified by flow cytometry via measurement of a decrease in EGFP fluorescence intensity. Cells were seeded in a 24-well plate and grown to 50% confluence. For the transfection, the growth medium in wells was replaced with the complete medium containing an MLNC bearing one of the siRNAs (As-OMe/S, As-OMe/S-F*, or As-OMe/S-OMe*). The quantity of the MLNC corresponded to 1.5 nM gold. After 4 h, the transfection was stopped, and the cells were washed with PBS and incubated in the fresh complete medium for 72 h. Next, the cells were washed with PBS, trypsinized, and resuspended in the complete medium. To detect the silencing effect of siRNA on EGFP synthesis, the cells were analyzed on a NovoCyte (ACEA Biosciences, San Diego, CA, USA) and in the NovoExpress software (ACEA Biosciences). In each experiment, at least 10,000 stand-alone events were acquired. For each type of MLNC, three independent experiments were conducted.

### 2.10. Statistical Analysis

Results of siRNA adsorption on AuNPs and the silencing effect were statistically evaluated by one-way ANOVA. Data with *p* ≤ 0.05 were considered statistically significant. Online ANOVA Calculator (https://www.statskingdom.com/180Anova1way.html accessed on 1 November 2022) was used for this analysis. Data are expressed as mean ± S.D. for at least three independent experiments.

## 3. Results and Discussion

As evidenced by numerous publications, chemical modifications of siRNA are a potent way to increase the efficiency and therapeutic potential of these nucleic acids. Modifications of siRNA ensure their chemical stability and resistance to nucleases as well as suppress the activation of an innate immune response and off-target toxicity [1,28,29,30]. Recent reviews examined various types of chemical modifications of siRNA and their effects [3,5,8,21]. As a part of nanocarriers, siRNA is usually shielded by components of the carrier, which solves some of the problems listed above, bringing to the forefront the problem of increasing the resistance of siRNA to intracellular nucleases and, accordingly, increasing siRNA efficiency. Taking into account that only 2–3% of the nanocarriers introduced into the bloodstream enter target cells [18,19,20], it is important to maximally load each carrier with siRNA molecules. Obviously, the number of siRNA molecules released from the carrier into the cytosol directly determines siRNA efficiency; therefore, we analyzed the number of siRNA molecules adsorbed on the carrier (AuNPs) as a function of chemical modification.

In this work, we used a previously generated library of siRNAs aimed at suppressing the synthesis of the GFP protein in the SC-1 cell line [25]. The sense and antisense strands of siRNAs contained modification 2′-OMe or 2′-F at nuclease-labile sites; for several compounds, these positions were additionally strengthened by the introduction of a PG modification into the internucleotide phosphate (Appendix A, adapted from [25]).

Previously [25], for all types of siRNA, a melting point, stability in the presence of RNase A, and silencing efficiency were determined. The introduction of the PG modification slightly reduced the thermal stability of siRNA but dramatically increased its resistance to RNase A, and the protective effect was observed not only for the modified site but also for neighboring ones. Because duplexes containing the PG modification in the antisense strand almost did not change the expression of the target protein (GFP), it is advisable to introduce this modification only into the sense strand of the duplex in order to study the biological effect [25]. On the basis of the series of obtained data, we investigated effects of the chemical modifications of siRNA on the adsorption of siRNA on AuNPs and on the biological activity of siRNA within an MLNC.

### 3.1. Efficiency of Adsorption of siRNA on AuNPs

Rational design of a modified siRNA can significantly increase the efficiency of siRNA and affect its delivery to a target [31]. In this regard, we assessed the influence of the modification of siRNA with PG on the efficiency of adsorption of each of the 25 modified siRNAs on the surface of AuNPs. To evaluate the effect of the chemical modifications of siRNA on physicochemical properties of the core particles, we introduced the [^32^P] label at the 5′ end of antisense strands of siRNAs; using this label, we determined the adsorption efficiency of the siRNA on the surface of AuNPs (Figure 2).

The results clearly indicate (Figure 2) a wide variation in the density of the coating of AuNPs by siRNA molecules: 11 to 66 molecules per AuNP. When siRNAs were analyzed that contain a native antisense strand (gray category, Figure 2), or an antisense strand containing 2′-OMe (red category), or 2′-OMe/PG (blue category), there was no dependence of the number of adsorbed siRNAs per AuNP on the type of sense strand used for the categories (Figure 2). At the same time, for siRNAs based on the antisense strand containing fluorine at the 2′ position (green category), adsorption efficiency depended on the type of modification of the sense strand. For instance, among siRNAs based on As-F, siRNA As-F/S-F* stands out, and among siRNAs based on As-F* (yellow category), siRNAs As-F*/S-F and As-F*/S-F* stand out (Figure 2). Analysis of the obtained data showed that the categories of siRNAs containing the modified antisense strand were statistically significantly different from the control category of siRNAs containing the native antisense strand (gray category, Figure 2).

The introduction of 2′-OMe units into the antisense strand led to an increase in the efficiency of siRNA adsorption on AuNPs (red category, Table 2). The introduction of a PG modification into 2′-OMe-modified antisense strands (blue category) in most cases caused a significant increase in the number of adsorbed siRNAs as compared to the red category (Table 2). The greatest increase in sorption efficiency was observed for siRNA As-OMe*/S-F*: the increase was 51% relative to a similar duplex that does not contain PG in the antisense strand of As-OMe/S-F*. A decrease in the number of negative charges because of the PG modification of the internucleotide phosphate should improve the adsorption of the siRNA on the polarized gold surface coated with citrate anions [32].

Nonetheless, when comparing the adsorption of siRNA As-OMe/S-OMe* with the adsorption of siRNAs As-OMe*/S-OMe*, As-OMe*/S-OMe, and As-OMe*/S-OMe*, we did not find statistically significant differences in the efficiency. Possibly, the lack of differences is related to how the above-mentioned modified siRNAs are stacked on the surface of AuNPs. The uniform distribution of 2′-OMe and PG modifications along the entire length of siRNA (Appendix A, adapted from [25]) ensures efficient adsorption of the entire duplex. It can be hypothesized that the modification of siRNAs enlarges the area of the siRNA contact with the surface of AuNPs as compared to unmodified siRNA. Apparently, the surface of AuNPs can be saturated with siRNA molecules, and for this reason there is no increase in the adsorption efficiency after 2′-OMe modification of siRNA, regardless of the presence of the PG modification in one or two siRNA strands.

Let us examine the combination of modifications where a fluorine atom is present at the 2′ position of the sense strand (blue category, Figure 2). For example, siRNAs As-OMe*/S-F* and As-OMe*/S-F are adsorbed with the same efficiency. At the same time, the adsorption efficiency of As-OMe*/S-F* is higher than that of As-OMe/S-F* (*p* < 0.05). In this case, we see a change in the number of adsorbed siRNA molecules on the surface of AuNPs after introduction of the PG modification into both strands. The introduction of fluorine into the 2′ position of the antisense strand leads in most cases to a decrease in the adsorption efficiency as compared to the duplexes based on the native antisense strand. siRNA As-F/S-F* shows minimal adsorption: 11 ± 2 siRNA molecules per AuNP. Perhaps this is due to the high electronegativity of fluorine, resulting in repulsion from the polarized gold surface, which carries a negative charge.

The introduction of the hydrophobic uncharged PG modification into As-F (yellow category, Figure 2) raises the efficiency of duplex adsorption relative to the green category. In the case of As-F*, no increase in adsorption was noted for siRNAs with a PG-modified sense strand as compared to siRNAs containing a PG-free sense strand, as is the case for As-OMe*.

It is noteworthy that the adsorption of siRNAs containing S-F or S-F* reached “record” 66 ± 7 and 66 ± 10 molecules of siRNA per one AuNP, respectively. Such an increase in the loading of AuNPs may also be due to how the siRNAs are arranged on the surface. It is possible that the siRNAs in which both strands contain a fluorine atom at the 2′ position have a smaller area of the contact with the surface than do similar siRNAs in which a 2′-methoxy group is present in one or both strands. An analysis of our results indicates that the efficiency of adsorption of siRNAs having the same sequence on AuNPs depends on the combination of the introduced modifications. Nonetheless, the anomalous behavior of siRNAs As-F/S-OMe* and As-F/S-F* suggests that the efficiency of siRNA adsorption on the surface of AuNPs is affected by other unknown factors, aside from the introduced modifications.

### 3.2. Stability of siRNA within AuNP-siRNA Cores

Next, we examined the stability of siRNA within the cores in the presence of FBS. Because siRNAs containing the PG modification in the antisense strand have only a weak biological effect [25], we did not test them.

To characterize the “behavior” of siRNA on the core in the presence of nucleases and phosphatases from FBS (which simulates the conditions inside the cell for a delivered siRNA [15]), the stability of siRNAs containing As, As-OMe, or As-F was analyzed in the presence of 10% of FBS. Previously, we have found that in a biological medium, siRNA molecules are desorbed from the surface of AuNPs (from the core) [15]. In 10% FBS, the degree of degradation of the desorbed siRNA varied from 10% to 44% (Figure 3).

### 3.3. The Efficiency of Delivery of siRNA as Part an MLNC

To determine the efficiency of delivery of siRNA (when packaged into an MLNC) into cells, an estimated amount of siRNA released from an MLNC in the cell was calculated (Table 3). Considering that in the cell cytosol, the siRNA is completely desorbed from the surface of AuNPs, we chose the following Formula (2):NsiRNA(delivered) = NsiRNA(adsorbed per AuNP) ∙ (100% − D)/100%(2)
where D is the degree of degradation of the siRNA desorbed from nanoparticles in 10% FBS; N—number of siRNA molecules.

The calculations revealed that a 1.5-fold-greater amount of modified siRNA relative to the native siRNA can be released in the case of three siRNA types: As-OMe/S, As-OMe/S-OMe*, and As-OMe/S-F*. It is these siRNAs that we investigated for the effectiveness of silencing after delivery within an MLNC to cells.

The MLNCs for the experiment on cells were prepared in accordance with our previously developed protocol [13]. All prepared samples of purified and concentrated MLNCs were examined by DLS analysis and TEM. The hydrodynamic characteristics of the obtained samples of MLNCs are given in Table 4.

The results of DLS analysis indicated that the use of As-OMe/S-F*-modified siRNA does not cause a significant change in the hydrodynamic dimensions of the MLNC as compared to the MLNC carrying native siRNA (As/S). By contrast, the inclusion of As-OMe/S or As-OMe/S-OMe* in an MLNC resulted in a significant reduction in hydrodynamic size.

TEM analysis of samples of MLNCs carrying different types of siRNA, after purification and concentration in accordance with ref. [13], revealed polymorphic particles containing one to ten AuNPs (Figure 4). It should be noted that all the obtained samples were almost free of “bare” cores and “empty” lipid particles. MLNCs containing several cores predominated in all the samples. There were no visually noticeable differences in structure among the MLNCs carrying various siRNAs.

By means of the effectiveness of downregulation of green fluorescent protein expression, we next determined the biological activity of selected siRNAs delivered to cells via the MLNC. Cultured HEK293FT EGFP cells stably expressing EGFP were treated with one of the MLNCs. During transfection, the particle concentration was 1.5 nM in terms of gold. The efficiency of silencing was determined by flow cytometry as a change in the intensity of cell fluorescence after transfection of cells with the MLNC. All cells treated with the MLNC manifested a decline of fluorescence intensity (Figure 5), suggesting suppression of the synthesis of EGFP and hence good delivery efficiency.

We have previously shown that siRNAsAs-OMe/S, As-OMe/S-F*, and As-OMe/S-OMe* when in complex with Lipofectamine 3000 all yield the same decrease in fluorescence intensity [25]. On the other hand, the same siRNAs within MLNCs showed different magnitudes of protein synthesis inhibition, as determined by the change in fluorescence of transfected cells (Figure 5).

The silencing efficiency for the MLNC carrying As-OMe/S is comparable to that for the MLNC carrying native siRNA (Figure 5). By contrast, transfection of cells with As-OMe/S-F* or As-OMe/S-OMe* led to a more pronounced suppression of EGFP gene expression (Figure 5). This dissimilarity may be due to differences in (a) the number of delivered siRNA molecules; (b) the efficiency of the release of siRNA from the MLNC, and (c) the stability of siRNA under intracellular conditions. As readers can see from the data in Table 3, cores carrying siRNA As-OMe/S, As-OMe/S-F*, or As-OMe/S-OMe* differ from the core carrying the native siRNA by a greater number of siRNA molecules per AuNP. PG-modified siRNAs (As-OMe/S-F* and As-OMe/S-OMe*) are adsorbed 1.5 times more efficiently on the surface of AuNPs as compared to the native siRNA. For As-OMe/S, the adsorption efficiency is 1.73 times higher; however, the MLNC containing this siRNA has lower efficiency compared to the MLNC containing As-OMe/S-F* or As-OMe/S-OMe*. Therefore, the advantage of As-OMe/S-F* and As-OMe/S-OMe* is due to the protective action of the PG modification.

Thus, MLNCs carrying siRNA As-OMe/S-F* or As-OMe/S-OMe* showed the highest efficiency of suppression of EGFP protein fluorescence in HEK293FT cells. This finding primarily means the effective protection of siRNA in the cytosol by the chemical modifications. Accordingly, the PG modification’s position (in the strand), which affects the number of adsorbed siRNA molecules, determines the specific effect of the entire nanocomplex.

The manifestation of specific effects by all the tested siRNAs in the HEK293FT cell line indicates the effective protective function of the lipid membrane in the obtained MLNCs; this membrane was able to preserve siRNA in a culture medium containing 10% of FBS, which successfully degrades siRNA because of the nucleases in it. The MLNC confirmed once again that it is a reliable carrier of siRNA, regardless of the presence of chemical modifications in siRNA strands.

## 4. Conclusions

siRNA is generally recognized as a means of specific suppression of protein synthesis in the cell, and this property determines the attractiveness of siRNA as a drug. The list of such drugs approved for clinical use includes five therapeutics, even though 20+ years have passed since the first publication of successful siRNA-driven silencing of mRNA in mammalian cells [33]. Physicochemical and biological properties of siRNA necessitate its (a) delivery to target cells and (b) protection from aggressive factors in immediate surroundings, including nucleases present in all biological fluids and inside the cell. In this work, we studied the effect of combined modifications, i.e., PG/2′-OMe or PG/2′-F in siRNA or of siRNA containing only 2′-OMe, and 2′-F substitutions, on the adsorption of siRNA molecules on AuNPs during the formation of the core of a nanocarrier (MLNC). The adsorption efficiency varied widely, depending on the type and combination of the modifications introduced into siRNA. The introduction of the PG modification into at least one of the strands raised the efficiency of siRNA adsorption on the surface of AuNPs in most cases. Our study shows that the efficiency of siRNA adsorption on the surface of AuNPs, together with the resistance of siRNA to nucleases, should be taken into account when one designs an siRNA carrier. It is these two parameters that make it possible to calculate the amount of siRNA released into the cytosol per carrier unit. For example, less stable derivatives of siRNA based on As-OMe (demonstrating more efficient adsorption) had a more pronounced biological effect as compared to the control siRNA.

Our main finding is that chemical modifications of siRNA affect its binding to an AuNP-based carrier, thereby directly affecting the efficiency of inhibition of the target protein expression. We believe that these results may be useful for the development of siRNA carriers based on other solid nanoparticles.

## Figures and Tables

**Figure 1 nanomaterials-12-04450-f001:**
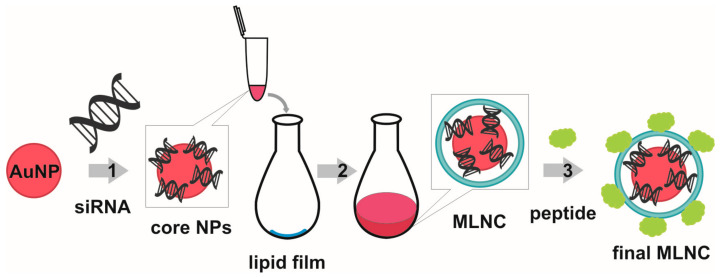
MLNC preparation scheme, basic stages: 1: non-covalent covering of AuNPs with siRNA (preparation of core NPs); 2: enveloping of the core NPs with pre-prepared lipid film (preparation of MLNC); 3: doping of lipid envelope with cell penetrating peptide (preparation of final MLNC). Adapted from [13].

**Figure 2 nanomaterials-12-04450-f002:**
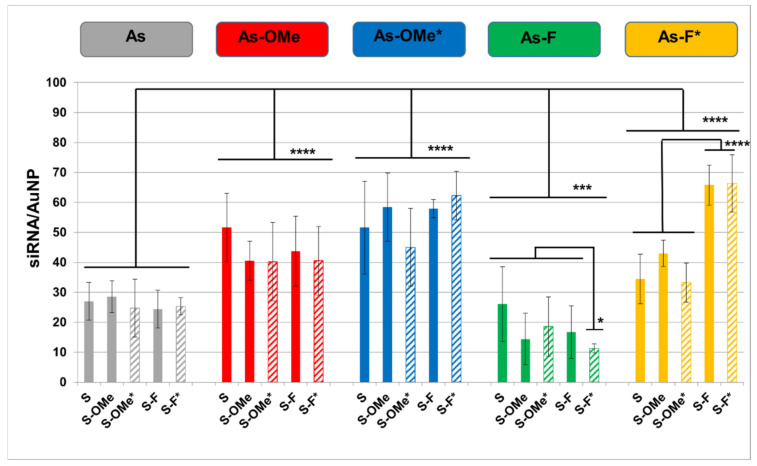
The influence of the chemical modifications on siRNA adsorption on AuNPs. siRNAs are categorized according to the type of their antisense strand; for clarity of presentation of the different categories, different colors are used. The boxes in the top row show categories according to the structure of the antisense strand (As) in siRNA. Colored vertical columns represent the data for each member of a category (each siRNA). Striped columns denote PG-modified siRNAs, and an asterisk (*) in an siRNA name means the PG-modified strand. The black lines combine sets of data that were compared with each other. Mean values (±SD) from at least four independent experiments are presented. The data were statistically assessed by one-way ANOVA. Significant differences between siRNA categories as well as siRNAs inside a category: * *p* < 0.05; *** *p* < 0.0005; **** *p* < 0.00005.

**Figure 3 nanomaterials-12-04450-f003:**
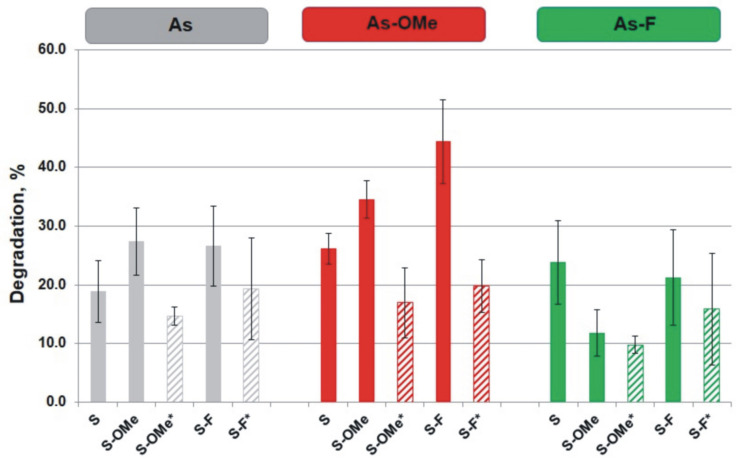
Degradation of the siRNA desorbed from AuNPs in FBS. siRNAs are categorized according to the type of their antisense strand. Boxes in the upper row show the category of the antisense strand (As) of an siRNA. For clarity of presentation of different categories of siRNAs, different colors are used. Colored vertical columns represent the data for each member of a category (each siRNA). Striped columns denote PG-modified siRNAs, and an asterisk (*) in an siRNA name indicates the PG-modified strand. Mean values (±SD) from at least four independent experiments are presented.

**Figure 4 nanomaterials-12-04450-f004:**
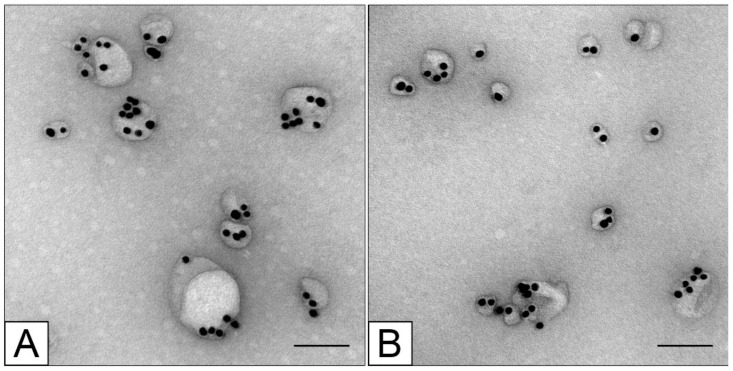
Ultrastructure of MLNCs carrying (**A**) unmodified or (**B**) modified siRNA (As-OMe/S). Negative staining with 0.5% uranyl acetate (TEM). Scale bars correspond to 100 nm.

**Figure 5 nanomaterials-12-04450-f005:**
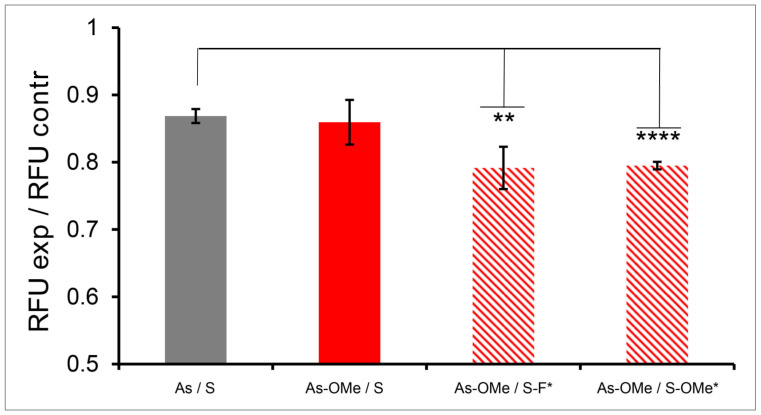
The normalized mean value of cell fluorescence (RFUexp/RFUcontr) after incubation of HEK293FT EGFP cells with one of MLNCs. Striped columns denote PG-modified siRNAs, and an asterisk (*) in an siRNA name means the PG-modified strand. Data were obtained by flow cytometry. At least 10,000 events were acquired in each sample; mean values (±SD) from three independent experiments are presented. The data were statistically assessed by one-way ANOVA. Significant differences between cell groups: ** *p* < 0.05; **** *p* < 0.0005. RFU: relative fluorescence units.

**Table 1 nanomaterials-12-04450-t001:** Sequences of single-stranded oligoribonucleotides (ssRNAs).

ssRNA		Sequence 5′→3′
sense	S	CAAGCUGACCCUGAAGUUCTT
	S-OMe	CmAAGCUmGACCCUmGAAGUUCTT
	S-OMe*	Cm*AAGCUm*GACCCUm*GAAGUUCTT
	S-F	CfAAGCUfGACCCUfGAAGUUCTT
	S-F*	Cf*AAGCUf*GACCCUf*GAAGUUCTT
antisense	As	GAACUUCAGGGUCAGCUUGTT
	As-OMe	GAACUUCmAGGGUCmAGCUUmGTT
	As-OMe*	GAACUUCm*AGGGUCm*AGCUUm*GTT
	As-F	GAACUUCfAGGGUCfAGCUUfGTT
	As-F*	GAACUUCf*AGGGUCf*AGCUUf*GTT

Nm: modification 2′-OMe, Nf: 2′-fluoro modification, and *: a phosphoryl guanidine modification.

**Table 2 nanomaterials-12-04450-t002:** A comparison of the adsorption of an siRNA containing the PG modification in the antisense strand with a similar siRNA without the modification of internucleotide phosphate in the antisense strand.

Antisense Strand Sense Strand	As-OMe Number of siRNA Molecules per 1 AuNP	As-OMe* Number of siRNA Molecules per 1 AuNP	The Increase in the Efficiency of siRNA Adsorption on AuNPs ^1^ %
S	52 ± 11	52 ± 16	0
S-OMe	41 ± 7	58 ± 11	41
S-OMe*	40 ± 13	45 ± 13	3
S-F	44 ± 12	58 ± 3	32
S-F*	41 ± 12	62 ± 8	51

* ^1^ the values in column 3 are calculated as a change (in %) in the ratio of values from column 2 to values from column 1. The color in the columns show siRNA groups according to the structure of the antisense strand (As) in siRNA.

**Table 3 nanomaterials-12-04450-t003:** Calculated numbers of released siRNA molecules in the cell from one MLNC containing one AuNP core.

Sense Strand	Antisense Strand	Number of Full-Size siRNA Molecules
S	As	22
S-OMe		21
S-OMe*		21
S-F		18
S-F*		20
S	As-OMe	**38**
S-OMe		27
S-OMe*		**33**
S-F		24
S-F*		**33**
S	As-F	20
S-OMe		13
S-OMe*		17
S-F		13
S-F*		10

The color in the columns show siRNA groups according to the structure of the antisense strand (As) in siRNA.

**Table 4 nanomaterials-12-04450-t004:** Physicochemical parameters of MLNCs carrying different siRNAs.

siRNA in MLNC	Mean Hydrodynamic Diameter (± SD), nm	PDI
As/S	328.7 ± 163.1	0.16
As-OMe/S	127.7 ± 57.2	0.2
As-OMe/S-F*	306.1 ± 113.5	0.13
As-OMe/S-OMe*	200.3 ± 83.9	0.16

## Data Availability

Data are available on request from the corresponding author.

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
