# Peer review of "Chemical Modifications Influence the Number of siRNA Molecules Adsorbed on Gold Nanoparticles and the Efficiency of Downregulation of a Target Protein"

_nanomaterials, 2022, doi:10.3390/nano12244450_

Round 1

Reviewer 1 Report

The authors have designed multilayer nanoconstructs (MLNCs) with AuNP core bearing different chemically modified siRNAs and compared the adsorption efficiency of siRNA on a single AuNPs. The AuNPs with modified siRNAs showed higher stability and increased biological activity. Although interesting results have been achieved, there are many mistakes in the manuscript and some conclusions are not well supported by the data. So, I recommend to you that major revision is needed before the manuscript is published. Corresponding comments are as follows:

1. There are too many mistakes in the manuscript. For example, “NsiRNA(sorbet per AuNP)” in line 402. I strongly recommend the authors reexam the manuscript seriously and professional editing needs to be performed.

2. It is better to draw Figure 1 with structures step-by-step.

3. For the conclusions shown in the third paragraph of “Results and Discussion”, corresponding data should be given. For example, the authors demonstrate that “The introduction of the PG modification slightly reduced thermal stability of siRNA but dramatically increased its resistance to RNase A, and the protective effect was observed not only for the modified site but also for neighboring ones.”, while there is not any data to support it.

4. As shown in Table 2, the number of siRNA molecules on AuNPs with As-OMe*/S-OMe* (45) was greatly decreased compared with As-OMe*/S-OMe (58). Does it mean the PG modification on sense strand influence the adsorption efficiency on AuNPs? Also, the number of the released siRNA from a single AuNPs with As-OMe/S was highest compared with those have modifications on sense strand as shown in Table 3. Does it mean that any modifications on sense strand would influence the desorption efficiency of siRNA on AuNPs?

5. The authors state in line 383 that “In 10% FBS, the degree of degradation of the desorbed duplex varied from 9.8% to 44.4% (Figure 3).”, while I cannot find any data shown degradation efficiency ranging from 9.8% to 44.4% in Figure 3.

6. The duplexes with a PG group showed comparable or less degradation efficiency of those without PG residues. Since there are the situations with comparable degradation efficiency, the conclusion that “the introduction of a PG residue into any of the sense strands protected the nucleic acid released from the surface of the AuNPs (line 387)” is not a precise conclusion.

7. The authors indicate that “D is the degree of degradation of the siRNA desorbed from nanoparticles in 10% FBS (Table 3).” in line 403, while the degradation efficiency was shown in Table 2.

8. It is better to change the number of siRNA on AuNPs to integers without decimal.

Author Response

Dear Reviewer, the answer to your Comments is in the attached file.

Reviewer 2 Report

Authors proposed the effective modifications of siRNA against GFP in this study. But it seems low advantage of the effect of siRNA knockdown, because  the results of Figure 5 showed that the  knockdown efficiency is too low (〜20%). I would like to know the additional experiments such as western blotting and RT-PCR. In addition, the another experiment are required to use other sequences of siRNA that already evaluated the knockdown efficiency.

Author Response

(The authors gave the same response as above.)

Reviewer 3 Report

In this work, chemical modifications that influence the number of siRNA molecules adsorbed on gold nanoparticles were discussed. However, the manuscript lacks a lot of key data and is not suitable for publication in this journal in its current state. The details are as follows:

(1) Lack of scale in Fig 4.

(2) Pay attention to the subscript of the full-text chemical formula and rewrite.

(3) Table 2, Numbers have no corresponding arithmetic unit

(4) Section 2.6, images not seen in the article

(5) Section 2.8, it is necessary to use images to prove that EGFP gene has successfully entered the host cell genome

Author Response

(The authors gave the same response as above.)

Reviewer 4 Report

Review comments are attached

Author Response

(The authors gave the same response as above.)

Round 2

Reviewer 1 Report

The authors have sincerely addressed most of the criticisms and revised manuscript is significantly improved. However, I still recommend the authors to redraw Figure 1 with structures step-by-step, although it has already showed in their previously paper.

Author Response

Dear Reviewer,
Thank you for your work with our manuscript and its positive evaluation.
We have tried to improve the scheme in Figure 1, and we really hope that the new version will satisfy you.
Sincerely, authors

Reviewer 3 Report

The manuscript has been modified according to the requirements, and the key issues concerned by the reviewer has been solved. Suggest being accepted.

Author Response

Dear Reviewer,
Thank you for your work with our manuscript and its positive evaluation.